# SWIPT-Pairing Mechanism for Channel-Aware Cooperative H-NOMA in 6G Terahertz Communications

**DOI:** 10.3390/s22166200

**Published:** 2022-08-18

**Authors:** Haider W. Oleiwi, Hamed Al-Raweshidy

**Affiliations:** Department of Electronic and Electrical Engineering, Brunel University, London UB8 3PH, UK

**Keywords:** 6G wireless communications, cooperative networking, energy/spectral efficiencies, energy harvesting, H-NOMA, outage probability, SWIPT-pairing, THz

## Abstract

The constraints of 5G communication systems compel further improvements to be compatible with 6G candidate technologies, especially to cope with the limited wavelengths of blockage-sensitive terahertz (THz) frequencies. In this paper integrating cooperative simultaneous wireless information and power transfer (SWIPT) and hybrid-non-orthogonal multiple access (H-NOMA) using THz frequency bands are suggested. We investigated and developed an optimal SWIPT-pairing mechanism for the multilateral proposed system that represents a considerable enhancement in energy/spectral efficiencies while improving the significant system specifications. Given the system performance investigation and the gains achieved, in this paper, wireless communication systems were optimized and upgraded, making use of promising technologies including H-NOMA and THz communications. This process aimed to alleviate the THz transmission challenges and improve wireless connectivity, resource availability, processing, robustness, capacity, user-fairness, and overall performance of communication networks. It thoroughly optimized the best H-NOMA pairing scheme for cell users. The conducted results showed how the proposed technique managed to improve energy and spectral efficiencies compared to the related work by more than 75%, in addition to the dynamism of the introduced mechanism. This system reduces the transceivers’ hardware and computational complexity while improving reliability and transmission rates, without the need for complex technologies, e.g., multi-input multi-output or reflecting services.

## 1. Introduction

Current wireless communication systems lack to meet the new requirements of the ever-updating next generations. This requires compulsory integration of the leading edge of promising technologies and intelligent applications to comply with distance-dependent terahertz (THz) constraints. It is essential to develop a suitable capable communication system to satisfy the expected features and demands, i.e., ubiquitous connectivity, supreme SE, minimal latency, huge data rate, system robustness, user fairness, supporting emergent applications, energy efficiency (EE), spectral efficiency (SE), and cost-effectiveness. Revolutionary research across the world has identified THz frequencies as the future of wireless communications [1,2,3]. EE and SE are pivotal factors to assess and enhance communication systems to satisfy the emergent essential 6G applications [4,5,6,7]. EE is an essential criterion in the next era of wireless communications and its overwhelmed infrastructure due to the rising power consumption of the required elements to connect a huge number of devices supporting the principle of the internet-of-everything (IoE) [8,9,10,11,12,13]. Hence, EE in 6G communications is mandatory to save energy and meet the practicality of 6G communication networking [14]. As a critical topic, terahertz communications (0.1–10 THz) has attracted great attention from the research community, playing a very important role in 6G and generations beyond. SE depends on the availability of bandwidth (BW) and working frequency. It is the backbone of the next wireless communications era given its valuable features as well as the various services that THz provides [15]. THz communication complements mmWave and optical bands as an alternative to fiber optics links of certain use cases, i.e., backhaul to backhaul/fronthaul, kiosk-users, data-centers internal links, internal device links, and THz-to-fiber links [16]. However, absorption and path losses effects of THz frequencies divide the spectra into spectral bands that are being explored to comply with 6G communication services [17]. The new generation’s ubiquitous coverage necessitates a revolutionary upgrade to the existing systems toward establishing robust and reliable wireless systems with extraordinary capabilities.

To this end, this paper is motivated by the capability of integrating the evolutionary technologies and developing a channel-aware path selection mechanism, targeting the establishment of a reliable and scalable simplified system by the enhancement of bandwidth-competitive 6G THz communications, and overcoming the constraints of the current resource-scarce systems. Moreover, this work optimized the existing communication systems by adopting hybrid-NOMA, THz with cooperative SWIPT technologies to comply with the intended goals, highlighting the attained gains. The paper’s contributions to knowledge are:(1)We designed a modified integrated (energy and spectral efficient) wireless communication system with powerful capabilities for the green communications era. It applies 6G candidate technologies to take full advantage of their characteristics for sufficient performance.(2)We utilized the practical application of hybrid-NOMA to the proposal for the utmost benefit of this scheme over single-carrier NOMA shortages.(3)We evaluated the best pairing strategy in H-NOMA to attain the best possible system performance.(4)We proposed a cost-effective simplified system, provided with a single input–single output high-directional antenna instead of other complexed schemes, to reduce computation and signal detection complexities in the receiver, maintaining sufficient SE, reducing power consumption, and increasing EE.(5)We investigated all the possible SWIPT pairs with the available (LOS) users to specify the best pair that provides the best performance.(6)We developed a dynamic mechanism to select the best SWIPT-pairing user out of all available users to guarantee fast and accurate dynamism.(7)We modified a scalable and upgradeable system while setting adjustable factors, i.e., coverage area, transmit power, carrier-frequency, bandwidth, and the simplest modulation.

The rest of the paper is structured as follows: Section 2 presents a brief background of the introduced system technologies. Section 3 compares a number of interesting previous works, explaining the intended objectives, followed by the paper’s contributions. Section 4 describes the proposal and addresses mathematical derivatives. It argues the idea behind proposing this system and the beneficial points. In Section 5 and Section 6, mathematical/simulation results are conducted and discussed, showing the outperformance of the proposed system over the state-of-the-art systems. In the end, we conclude this paper in Section 7.

## 2. Background

To meet the potential technical specifications, non-orthogonal multiple access (NOMA) is a strong candidate to be integrated with the 6G paradigm [18]. It enables the evolution of SE, outperforming the earlier strategies of orthogonal multiple access (OMA) in terms of SE, channel capacity, resource management, user-fairness, massive connectivity, and lower latency. The main procedure of NOMA is to carry out superposition coding combining users’ signals at the transmission end (Tx), where it must be realized and treated using successive interference cancellation (SIC) as a multi-user detection at the receiving end (Rx) to discard any interference. It enlarges the channel capacity, which relies on the channel bandwidth. This paper concentrates specifically on hybrid-NOMA (H-NOMA). Based on NOMA fundamentals, various power coefficients are allocated to the users based on their channel state information (CSI) and multiplex them in the power domain. To improve SE and eliminate complexity, user-clustering is important in THz NOMA communications based on users’ locations or other metrics. However, to reduce the complexity of clusters in some cases, only one main user’s CSI might be set as a reference for the remaining number of users within the NOMA cluster [18]. In the SISO-NOMA scheme, both BS and users are equipped with a single antenna that relies on CSI for users’ sorting, preparing them to implement better SIC at the receiving end. The larger the channel condition differences among users, the better NOMA performance we obtain [19]. H-NOMA is an integration of NOMA/OMA techniques. It is proposed to overcome the challenges or limitations that undermine the performance of those systems, e.g., the complexity and possibly of interference due to the huge number of users. The strategy of CSI acquisition is very important to determine the procedure and sequence of SIC; however, SIC is not CSI-based only, as there are QoS-based, hybrid-based, or other procedural SICs based on other strategies [20]. Due to the lack of transmission distance and the losses in THz communications, it is recommended to adapt NOMA-assisted cooperative networking to tackle those problems. Cooperative network relaying offers more reliability and capacity, enhancing the overall performance, especially when integrated with other modern technologies such as NOMA. NOMA’s SE can be improved using cooperative networking for better support to the blocked users or users with a weak signal-to-noise ratio (SNR) with THz communication, especially with the merging energy harvesting (EH) technique with cooperative NOMA [21], which the paper studies with THz frequencies. Reasonably, using cooperative networks will cause battery drainage of relaying of the user’s device. In addition, the terahertz-NOMA system experiences the burden of high computations of SIC at Rx; therefore, to comply with that problem, it is logically recommended to apply the EH technique [22]. Applying EH will exploit the radio frequency (RF) signals’ energy that surrounds most of the devices, e.g., energy belonging to other destinations. Using EH enables the relaying user to harvest that energy to use it again to retransmit the targeted user’s signal. In power splitting-based EH, the relaying user splits the received signal’s power into an EH partition (ψ) and an information-decoding partition (1 − ψ) to carry out EH and information-decoding at the exact time, i.e., simultaneous wireless information and power transfer (SWIPT) [23], as demonstrated in Figure 1. SWIPT enhances system capacity, outage probability (OP), and accordingly EE. It represents one of the potential technologies for physical layer optimization [24].

## 3. Related Works

In recent years, cooperative networking has been studied in several cases and scenarios, and its fruitful use has already been demonstrated. There are some disadvantages of using cooperative networks stated with wireless communications. Integrating the NOMA scheme and THz communications to attain a positive impact on wireless communications was explored thoroughly, showing the impairments, weaknesses, limitations, and the shortage of performance in academic and industrial environments. THz communications of [25], despite using intelligent reflecting surfaces (IRS) to improve the transmission between Tx and Rx, had to outperform the cooperative networks by all the means, simplicity, EE, and cost, not only optimizing reflectors but also to consider all other criteria, e.g., to minimize power consumption and to maximize EE. In [23], the authors presented important points, considering disadvantages of IRS design and deployment as a new technique to support 6G infrastructure, i.e., controlling instantaneous beam steering, interference, and EE. It also addressed the challenges of IRS deployment, i.e., (1) design and control joint communications components, required procedures and analyses, hardware impacts, on performance, total cost, required space for deployment, and the continuous maintenance; (2) potential failure influences because of environment or accidents; (3) IRS interactivity with the transmission instantaneous changes; and (4) IRS complexity as an additional computational burden. The authors in [26] explored grouping and pre-coding with energy optimization. They suggested a different way of user-clustering of THz multiple-input multiple-output (MIMO)-NOMA design by building an algorithm of artificial intelligence. In [27], the author proposed MIMO-based spatially multiplexed work by adopting a new index-modulation scale. He studied the MIMO technique to build ultra-throughput/SE systems. In [28], the energy allocation problem was studied with cooperative half duplex/full duplex MIMO-NOMA with THz for maximizing the data rates of users.

In the author’s recent work [1,2], similar integrated systems were proposed for SISO-NOMA and MIMO-NOMA, respectively. However, those papers did not consider channel awareness to propose a dynamic channel selection mechanism for a SWIPT-pairing near user, whereas the work in [3] studied SISO-H-NOMA from a clustering point of view without presenting a suitable mechanism to fulfill the gap.

Moreover, in a recent paper [29], the author investigated the role of multihoming and other essential enabling techniques in improving the performance of communication systems, whereas in a previous article [30], we looked into the influence of the multihoming concept on system reliability, efficiency, and performance, focusing on the importance of the multihoming strategy in maintaining communication of the multihomed users.

Based on the authors’ knowledge, the THz-based modified adaptable system has not been proposed yet concerning the cost/complex-to-performance trade-off. There have not yet been similar simplified and feasible mechanisms presented for optimum performance.

## 4. Methodology

This section is divided into two sub-sections; the first section studies hybrid-NOMA-pairing possibilities and optimizes the optimal pair to the associated users within the cell, whereas the second section studies the best SWIPT-pairing for the farthest blocked user (or distant cell-edge users). The main system model is shown in Figure 2; hence, some of the mathematical derivations were developed from our previous analysis in [1,2,3]. Table 1 describes the symbols of all the mathematical equations.

The transmission is considered over Rayleigh channel (mean = 0, variance = ${\mathrm{Transmission}\;\mathrm{Distance}}^{-\mathrm{THz}\;\mathrm{losses}}$) with additive white Gaussian noise (AWGN) (mean = 0, variance = $\sigma^{2}$. The probability density function (PDF) of a certain point is given by:(1)$$f_{\left(z;\sigma \right)}=\frac{1}{\sqrt{2\pi \sigma^{2}}}exp\left(-\frac{z^{2}}{2\sigma^{2}}\right)$$

### 4.1. The Best Hybrid-NOMA Strategy

In this sub-section, we studied the impact of the user-pairing scheme in NOMA–OMA multiple access (Figure 3) to be adopted for the next SWIPT-pairing sub-section.

#### 4.1.1. Near–Far Pairing (N–F)

With the N–F strategy, the nearest user to the BS (User4) pairs with the farthest user (User1). BS’s nearer user (User3) pairs with the second farther user (User2). Thus, in this strategy, User4 and User1 are paired in the first block, whereas User3 and User2 are paired in the second block. 

Within the first pair, User4 represents the near user (NU), while User1 is the far user (FU); however, power coefficients are required to be allocated by setting α4 < α1. Thus, User4 requires SIC implementation before detecting its intended signal, while User1 decodes its intended signal directly without SIC. Within the second pair, User3 represents NU, whereas User2 represents the FU; however, power coefficients must be allocated by setting α3 < α2. Thus, User3 implements SIC, whereas User2 detects its intended signal directly.

In pair 1, the rates of the users are given by:(2)$$R4,nf=\frac{1}{2}{log}_{2}\left(1+\frac{P\alpha 4{\left|h4\right|}^{2}}{\sigma^{2}\;}\right)$$
(3)$$R1,nf=\frac{1}{2}{log}_{2}\left(1+\frac{P\alpha 1{\left|h1\right|}^{2}}{P\alpha 4{\left|h1\right|}^{2}+\sigma^{2}\;}\right)\;$$

Similarly, for the second pair:(4)$$R3,nf=\frac{1}{2}{log}_{2}\left(1+\frac{P\alpha 3{\left|h3\right|}^{2}}{\sigma^{2}\;}\right)\;$$

Hence, *P* denotes transmit power, *αn* and *αf* are power coefficients of the near user and far user, respectively, and xn and xf denote the signals of the near user and far user, respectively.

The far user is not able to detect its signal because of the blockage; however, the signal of the near user is:(5)$$R2,nf=\frac{1}{2}{log}_{2}\left(1+\frac{P\alpha 2{\left|h2\right|}^{2}}{P\alpha 3{\left|h2\right|}^{2}+\sigma^{2}\;}\right)$$

NU’s rate is given by:*Rnf* = *R*1,*nf* + *R*2,*nf* + *R*3,*nf* + *R*4,*nf*(6)

#### 4.1.2. Near–Near, Far–Far Pairing (N–N, F–F)

The N–N, F–F strategy addresses that User4 pairs to User3, whereas User2 pairs to User1. This strategy pairs User4 to User3 and User2 to User1 for the two blocks.

In this strategy, U4 is NOMA NU as compared to U3. Therefore, we must allocate α4 < α3. User4 requires SIC implementation before detecting its intended signal, while User3 decodes its intended signal directly without SIC. In the other block, U2 is NOMA NU as compared to U1. Accordingly, we must allocate α2 < α1. Thus, User2 performs SIC and User1 decodes directly.

The rates in the first block are given by: (7)$$R4,nn=\frac{1}{2}{log}_{2}\left(1+\frac{P\alpha 4{\left|h4\right|}^{2}}{\sigma^{2}\;}\right)$$
(8)$$R3,nn=\frac{1}{2}{log}_{2}\left(1+\frac{P\alpha 3{\left|h3\right|}^{2}}{P\alpha 4{\left|h3\right|}^{2}+\sigma^{2}\;}\right)$$

Similarly, for the second pair:(9)$$R2,nn=\frac{1}{2}{log}_{2}\left(1+\frac{P\alpha 2{\left|h2\right|}^{2}}{\sigma^{2}\;}\right)$$
(10)$$R1,nn=\frac{1}{2}{log}_{2}\left(1+\frac{P\alpha 1{\left|h1\right|}^{2}}{P\alpha 2{\left|h1\right|}^{2}+\sigma^{2}\;}\right)$$

The NU rate is given by:*Rnn* = *R*1,*nn* + *R*2,*nn + R*3,*nn* + *R*4,*nn*(11)

### 4.2. The Best SWIPT-Pairing Mechanism

In this sub-section, we study SWIPT-pairing with three scenarios of pairing using one of the three available LOS near users (U4, U3, or U2) to be the DF relay pairing user with the targeted far user (U1), and then we develop a suitable mechanism to select the best SWIPT-pairing user using the minimum (min) function to find the lowest (nearest) channel fading of the available LOS users to be paired with the blocked farthest user to achieve the highest channel gain difference for the optimal NOMA pair. Otherwise, the mechanism selects the second-lowest fading (second-near) user, and so on.

Based on the Rayleigh fading equation, we propose a mechanism of selecting the best pairing user to act as a relay to user1, and we set the minimal Rayleigh fading channel
(12)$$R=\min\{H^{k}\}\;$$
where R is the selected relaying user, and *H* is a vector of Rayleigh channels for k-number of users; however, we propose *k* = 4.

Based on NOMA principles, the selected user is the NU (DF relay user), whereas the blocked user is the FU.

In the proposed system model, as we have 3 LOS available users with FU, the NU selection mechanism is expressed asNU = min{*h*2,*h*3,*h*4}(13)
where *h*2, *h*3, and *h*4 denote the Rayleigh fading channels of user2, user3, and user4, respectively, (each with its distance with the BS).

The proposed mechanism (shown in Figure 4) works in spacious open areas such as rural territories or the countryside (i.e., it is not possible to deploy other network equipment to aid THz communications). The system considers a downlink THz NOMA-based single-cell serving 4 users, where all the parties are provided with single highly directed antennas. The transmitter party (BS) combines users’ signals to broadcast them to the receivers (i.e., various channel-conditioned users) including the paired user (NU and FU), where we assume there is an existing obstacle blocking the BS-U1 link. Accordingly, U1 cannot receive the signal efficiently. The potential NU (U4, U3, or U2) has a sufficient channel gain. Based on NOMA, the signal of the FU is decoded and canceled by the NU implementing SIC before decoding the NU information signal. It is worth noticing that NU receives and decodes FU’s signal. Therefore, NU could aid FU’s connection as a DF-relay. To this end, the NU’s device power does not suffice to retransmit FU’s signal. Thus, we suggest that NU performs SWIPT to harvest energy from radio-frequency energy surrounding it. The communication process is carried out in two phases; the sender’s combined signal is received by NU in the initial phase. By adopting the power splitting technique, a portion of the received power will be captured, and the remaining power will be used for information decoding. The captured power is then used by NU to retransmit the signal to the FU.

The proposed system takes into consideration THz frequency characteristics and losses. Transmission link loss in the non-line-of-site path (NLoS) is much more than link loss in line-of-site (LOS); thus, the NLoS impact could be neglected when LOS governs [1]. In this paper, THz losses (*η*) are presumed to be very high. The channel gain can be calculated as:(14)$$h_{k}=\sqrt{A}\sqrt{\frac{1}{\;\eta \;}\;G}$$

Hence, *A* denotes antenna number, *G* refers to antenna gain, and *η* denotes THz source-to-destination losses, given by:(15)$$\eta ={\left(\frac{4\pi fd}{C}\right)}^{2}e^{a\left(f\right)d}$$

Hence, *f* denotes frequency, *d* denotes source-to-destination distance, *a*(*f*) denotes absorption coefficient, and c denotes light speed.

The derived closed-form according to the proposed scenario is:

Stage 1: The transmission of the superimposed signal in the first stage is shown as
(16)$$X=\sqrt{P}(\sqrt{\alpha n\;}xn+\sqrt{\alpha f\;}xf$$
where P denotes transmit power, *αn* and *αf* are the power coefficient of NU and FU, respectively, and *xn* and *xf* are the power signal of NU and FU, respectively. Due to the blockage, FU is not able to receive its signal, whereas the received signal at the NU is given by
(17)$$yn=\sqrt{P}\left(\sqrt{\alpha n\;}xn+\sqrt{\alpha f\;}xf\right)hsn+wn$$

Hence, *hsn* represents the BS-to-NU Rayleigh channel (mean = 0, variance = ${\mathrm{dsn}}^{-\eta }$), dsn denotes the BS-to-NU transmission distance, and *wn* refers to AWGN (mean = 0, variance = $\sigma^{2}$). Out of *yn*, NU extracts a portion of power as the EH coefficient (ψ). The rest of the energy (1 − ψ) is allocated to decode its data, which is represented by:(18)$$\begin{array}{l} yD=\left(\sqrt{\left(1-\psi \right)}\right)yn+weh\\ \;\;\;\;\;\;\;=\left(\sqrt{\left(1-\psi \right)}\right)\;\sqrt{P}\;\left(\sqrt{\alpha n\;}xn+\sqrt{\alpha f\;}xf\right)+\left(\sqrt{\left(1-\psi \right)}\right)\;wn+weh \end{array}$$

Hence, *weh* refers to thermal noise (zero mean, variance = $\sigma^{2}$). The tiny EH value of *wn* can be neglected; thereby *y*D will be:(19)$$yD=\left(\sqrt{\left(1-\psi \right)}\right)\sqrt{P}\;\left(\sqrt{\alpha n\;}xn+\sqrt{\alpha f\;}xf\right)+weh$$

From (17), NU decodes *xf* directly first. The achievable rate of the decoded FU’s data by the NU is
(20)$$Rnf=\frac{1}{2}{log}_{2}\left(1+\frac{(1-\psi )P\;\alpha f\;{\left|hsn\right|}^{2}}{(1-\psi )P\;\alpha n{\left|hsn\right|}^{2}+\sigma^{2}}\right)$$

By implementing SIC, NU’s rate is given by:(21)$$Rnf=\frac{1}{2}{log}_{2}\left(1+\frac{(1-\psi )P\;\alpha n\;{\left|hsn\right|}^{2}}{\sigma^{2}}\right)$$

The harvested energy during phase 1 is represented by:(22)$$PH=P\;{\left|hsn\right|}^{2}\;\zeta \;\psi$$

Hence, ζ denotes the electronic circuits’ EH efficiency.

Phase 2: In the next phase, by allocating the harvested energy (PH), NU retransmits the data meant for FU. Consequently, NU’s sent signal is:(23)$$\sqrt{\mathrm{PH}}\tilde{xf}$$

Accordingly, the signal at FU is given by:(24)$$\sqrt{\mathrm{PH}}\tilde{xf}hnf+wf$$
where *hnf* denotes the NU–FU Rayleigh fading channel. The achievable rate at the FU is
(25)$$Rf=\frac{1}{2}{log}_{2}\left(1+\frac{PH{\left|hsn\right|}^{2}}{\sigma^{2}}\right)$$

In order to evaluate the ideal value of the EH-coefficient, NU requires FU information decoding. Next, it can effectively convey the FU signal. Thus, the constraint Rnf > Rf∗ is set. 

Hence, Rf∗ represents the targeted rate of FU. Addressing that the NU rate to decode the FU signal must be greater than that of F. Rnf in (21) with the set condition is swapped to produce ψ
(26)$$\frac{1}{2}{log}_{2}\left(1+\frac{(1-\psi )P\;\alpha f\;{\left|hsn\right|}^{2}}{(1-\psi )P\;\alpha n{\left|hsn\right|}^{2}+\sigma^{2}}\right)>\mathrm{Rf}*$$
(27)$${log}_{2}\left(1+\frac{(1-\psi )P\;\alpha f\;{\left|hsn\right|}^{2}}{(1-\psi )P\;\alpha n{\left|hsn\right|}^{2}+\sigma^{2}}\right)>2\mathrm{Rf}*$$
(28)$$\frac{(1-\psi )P\;\alpha f\;{\left|hsn\right|}^{2}}{(1-\psi )P\;\alpha n{\left|hsn\right|}^{2}+\sigma^{2}}>2^{2\mathrm{Rf}*}-1$$

We denote $2^{2\mathrm{Rf}*}-1$ to be *τf*, representing the targeted value of FU’s signal to interference plus the noise ratio.
(29)$$\frac{(1-\psi )P\;\alpha f\;{\left|hsn\right|}^{2}}{(1-\psi )P\;\alpha n{\left|hsn\right|}^{2}+\sigma^{2}}>\tau f$$
(30)$$\left(1-\psi \right)P\;\alpha f\;{\left|hsn\right|}^{2}>\tau f(1-\psi )P\;\alpha n{\left|hsn\right|}^{2}+\tau f\sigma^{2}$$
(31)$$\left(1-\psi \right)P\;\alpha f\;{\left|hsn\right|}^{2}-\tau f(1-\psi )P\;\alpha n{\left|hsn\right|}^{2}>\tau f\sigma^{2}$$
(32)$$\left(1-\psi \right)P\;{\left|hsn\right|}^{2}\;(\alpha f-\tau f\alpha n)>\tau f\sigma^{2}$$
(33)$$\psi <1-\frac{\tau f\sigma^{2}}{P\;{\left|hsn\right|}^{2}\;(\alpha f-\tau f\alpha n)}$$

Verifying the constraint of ψ in (33), the equation is reformed as:(34)$$\psi =1-\frac{\tau f\sigma^{2}}{P\;{\left|hsn\right|}^{2}\;(\alpha f-\tau f\alpha n)}-\delta$$

Hence, *δ* refers to a very small value of $10^{-6}$; however, ψ represents the energy needed to decode information to attain the intended rate of FU.

The outage probability (OP) is the possibility of the data rate of the user falling below the targeted value. We assume Rn∗/Rf∗ as the target rates of NU/FU, respectively.

FU is in OP when the Rf rate of (25) falls underneath the targeted value, given by:PFU = Pr(Rf < Rf∗)(35)

NU decodes both signals accurately. Hence, both targeted levels must be equal to or greater than that of NU. By performing SIC, NU faces OP when both values in (21) and (25) do not meet that value of NU, mathematically given by:PNU = Pr(RNF < Rf∗) + Pr(RNF > Rf∗, Rn < Rn∗)(36)

Based on the IEEE standard in [31], and according to the available terahertz spectral bands, ref. [32] divided the gap into certain channels and bandwidths (approved globally), allocated BW depending on the system compatibility, application requirements, hardware limitation, and transmission conditions. 

In THz-SC PHY, BPSK and QPSK modulation schemes are mandatory; thus, we adopt the simplest scheme (BPSK) to improve system performance and mitigate the complexity, as increasing the modulation index, i.e., signal levels, leads to a greater bit error rate (BER) in addition to increasing the processing time and latency (this work achieves higher SE depending on the integrated technologies’ capabilities without the need for high-order modulation schemes). Moreover, the planner must utilize coverage distance with system performance to gain the trade-off between range and rate [31].

### 4.3. Optimal SWIPT-Pairing

The mechanism of the best SWIPT-pairing adopts the principle of the best channel-conditions difference between the targeted blocked user and the paired user. A higher distance between them causes a higher channel condition difference, which means better NOMA performance, especially when the available relaying NU locates at the closest point with regards to Tx. To gain an ideal channel difference, the farthest targeted user requires pairing with the nearest user to the BS, which is preferred to SWIPT; if that is not achievable, then the second nearest user to the BS must be tried, and so on. Then for the next pairing, the targeted user does the same procedure to find its pair starting from the nearest possible user to BS and farthest from itself. The blocked user will find the best available user for the optimal cooperative SWIPT scenario.

## 5. Implementation Environment

This work was simulated as the following (parameters of [30]): Firstly, we simulated and compared the performance with mathematical analysis to demonstrate the viability of employing H-NOMA to replace NOMA to achieve the intended objectives in order to gain the most advantages in comparison with NOMA and OMA. After that, we evaluated the paper’s core case to examine the three potential SWIPT pairs to the obstructed user (User1) and analyze each pair’s performance using parameter settings for frequency, bandwidth, transmit power, and transmission distance. Then we developed a suitable mechanism to select the optimal DF relaying user (NU) to the intended U1’s pairing. The system was simulated using MATLAB. Table 2 shows the simulation parameters.

We investigated system performance based on the influential THz factors. The suggested technique should make it possible for the obstructed user to continue communicating while being shadowed, failing to connect to BS. The ability to regulate THz shortfalls was then explored, demonstrating how this could enhance SE, EE, stability, and the entire efficiency. The modeling findings supported the optimized system’s obtained closed form and was compared to that of previous work. Table 3 describes hybrid-NOMA strategies.

## 6. Results and Discussion

The system simulation and analysis were carried out to prove the validity of the enhanced achievable rates and outage probability. All possible SWIPT pairs were studied.

### 6.1. H-NOMA, NOMA, and OMA Performance Comparison

The simulation of H-NOMA strategies was carried out in comparison to traditional single-carrier NOMA (SC-NOMA) and TDMA (Figure 5) in order to verify the rationale for selecting the optimal scheme among the MA techniques according to Figure 3 and Table 2. 

Based on Figure 5, whenever the channel-condition divergence of NOMA users is distinctive, we can observe that the (N–F) approach works better than other methods in some circumstances, significantly with THz, confirming that we can fully utilize NOMA in those situations. NOMA still outperforms TDMA with the (N–N, F–F) method, although not significantly better. Due to interference brought on by too many users using the same carrier, SC-NOMA’s performance is still insufficient. This leads to problems with complexity and interference. Consequently, employing a single carrier while increasing the number of people served is not recommended.

### 6.2. Proposal Simulation

To analyze system performance and compare it to the related work, and to show how this proposal presents an important improvement to wireless communications by utilizing the key-enabling techniques, we simulated the proposal for each potential relaying-user through 3 sections.

Figure 6 shows the average achievable rate of near and far users for the three possible SWIPT pairs, namely, U2–U1 (a), U3–U1 (b), and U4–U1 (c). We noticed that as a result of the EH mechanism, which utilized only the necessary power to achieve the required rate, capturing all the remaining power, NU peaked at 1 Gpbs/Hz. Increasingly, in (a), (b), and more in (c), NU achieved a better data rate, exceeding the target for the same power splitting ratio we used because of the lower distance (better SINR) from the BS, achieving better sum-throughput and SE accordingly. The FU rate increased without affecting NU stability by using the available harvested power in all cases; however, we could make use of the overused FU available energy for more EH operations. The best overall performance resulted in the U4–U1 SWIPT pair (c).

#### 6.2.1. OP versus Transmission Power

We notice from Figure 7 that FU showed higher OP than that of NU in all cases, although having higher rates of FUs as compared with NU. The best overall performance resulted in the U4–U1 SWIPT pair (c) as compared to those of (a) and (b) due to the greater channel difference between the NOMA near and far users (preferable). This supports the idea behind this work to assist the distant, the weak-conditioned, and the path-blocked users.

#### 6.2.2. Instantaneous-Rate versus Channel-Realization

To study the performance of the SWIPT pairs accurately, we examined the instantaneous rates under channel realization.

In Figure 8, we still observe that FU did not show better stability compared to NU in all the cases despite the larger power and data rate; FU’s instantaneous rate was gaining the same level as that in Figure 6 and some variations below the targeted level, which explains the variety of users’ performance. The best overall performance resulted in the U4–U1 SWIPT pair (c).

It is worth noting that the proposed system delivered a remarkable enhancement over that of [21] using simpler requirements and lower power/cost, and it gained higher SE and EE. It achieved better performance, showing the importance of EH and the emerged techniques.

### 6.3. Optimal SWIPT-Pairing Mechanism

In this section, we simulated the system model using the mechanism in (12) to select the best SWIPT partner to act as a DF relay to U1 out of the available LOS users, using the same parameters as in Section 4 and Section 5.

#### 6.3.1. Average Rate versus Transmission Power

Figure 9 illustrates how the aforementioned mechanism ran the best SWIPT-pair with the best performance, resulting in U4–U1 pair selection. This describes the importance of the channel condition difference between the paired NOMA users (NU and FU); the greater the channel difference, the better the NOMA performance. The privilege of selection dynamism was obtained by the proposed mechanism.

#### 6.3.2. OP versus Transmission Power

Similarly, Figure 10 shows how the same mechanism managed to select the best SWIPT pair with the best outage probability, resulting in the selection of the U4–U1 pair. It improved overall system reliability and OP by aiding the targeted user to opt for the best NU to pair with in terms of NU location and, accordingly, channel condition difference.

#### 6.3.3. Instantaneous Achievable Rate versus Channel Realization

To make sure that the proposed mechanism achieves accurately the best performance for users, Figure 11 above illustrates the simulation of the instantaneous achievable rates under channel realization. Once again, it is clearly shown that the proposed mechanism leads to the best user and overall performance accordingly by selecting the best pair to achieve the intended goal.

### 6.4. System Numerical Analysis and Simulation

We compared the proposal’s sum throughput and OP using simulations and analysis to prove the validity of the closed-form of the system model, leading to the preplanned objectives, utilizing similar input parameters, and setting a transmission power of 20 dBm.

#### 6.4.1. Sum Throughput versus Transmission Power

Based on the same parameters used to examine the validity of mathematical derivations, Figure 12 shows considerable matching with the system simulation. 

#### 6.4.2. User OP versus Transmission Power

Similarly, Figure 13 verifies the feasibility of mathematical analysis by conducting a point-to-point comparison and setting the exact parameters. It shows noticeable matching.

Remarkably, Figure 12 and Figure 13 depict clear consistency of both results, which confirm the accuracy of the analyses and system model validity of achieving the objectivity and novelty of the proven added values.

## 7. Conclusions

To meet the 6G stringent specifications, this work was carried out to fulfill the missing gap of 5G research and standardization, emphasizing the THz communication challenges. The optimal THz H-NOMA pairing approach for each user serviced was first examined in this paper. The outcomes demonstrated a substantial approach to overcoming SC-NOMA and THz limitations in order to accomplish the planned goals and adapt to the prospective wireless systems of the following generations. Then, a thorough analysis of all potential SWIPT-pairing prospective users of the cooperative THz H-NOMA was conducted. The closest user to the BS that offered the best system performance among all the available users was practically demonstrated to be an efficient SWIPT DF-relaying user. Moreover, the paper stated the significance of proposing such a mechanism, examining the feasibility of its management to opt for the optimal pair depending on users’ locations and SINR to develop an efficient system, achieving the best overall performance, e.g., maximum achievable EE/SE, using the merged technologies. It showed an improvement in overall performance when compared to the systems in use today (EE/SE are improved by 75 percent) in addition to the dynamism of the introduced mechanism, incorporating all the comparison criteria. The overall results proved the validity of the proposed techniques to maintain the reliability of ongoing THz communications, and the leverage of the developed mechanism to improve the system performance and precision almost perfectly. Finally, we examined the accuracy of numerical and simulation results that showed perfect matching. Further work must be done to automate the computational and procedural operations using artificial intelligence algorithms.

## Figures and Tables

**Figure 1 sensors-22-06200-f001:**
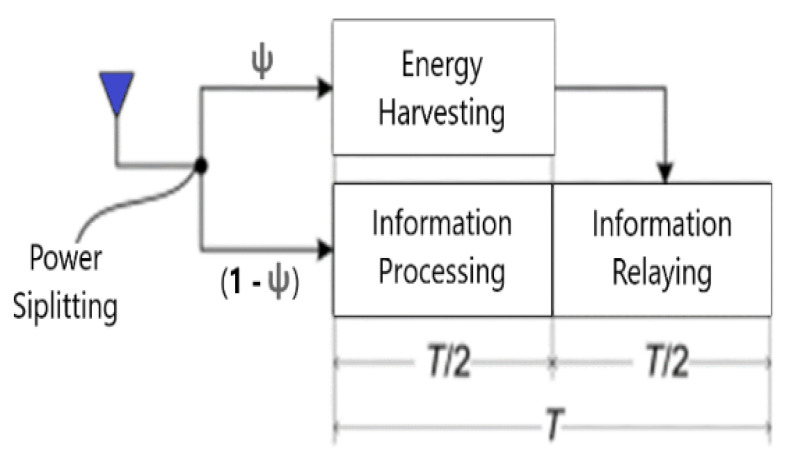
SWIPT with DF-relayed NOMA.

**Figure 2 sensors-22-06200-f002:**
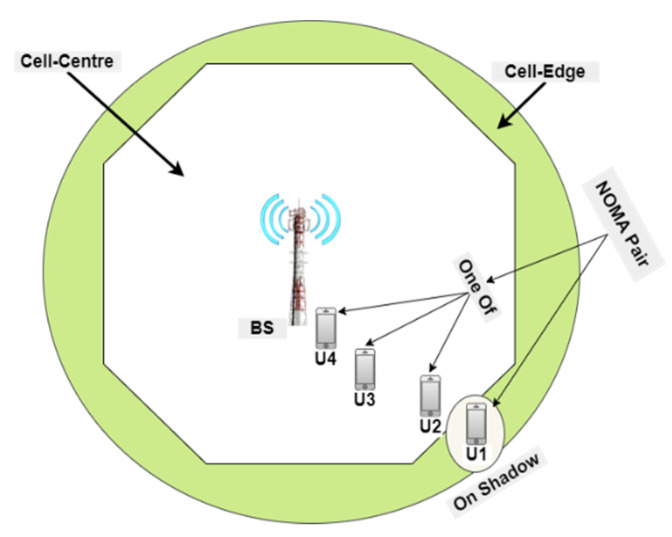
Main system model.

**Figure 3 sensors-22-06200-f003:**
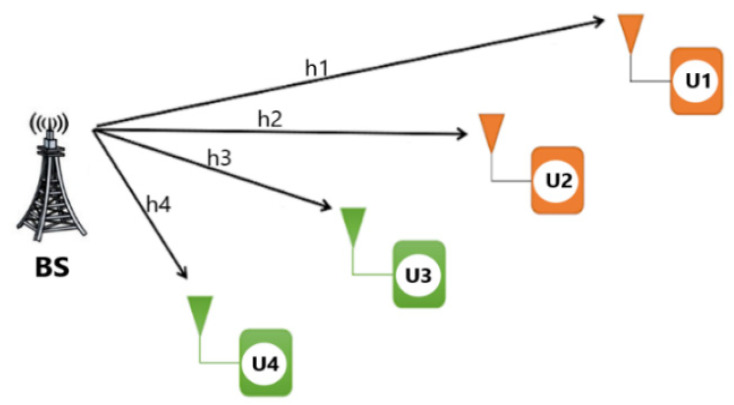
Hybrid-NOMA pairing strategy.

**Figure 4 sensors-22-06200-f004:**
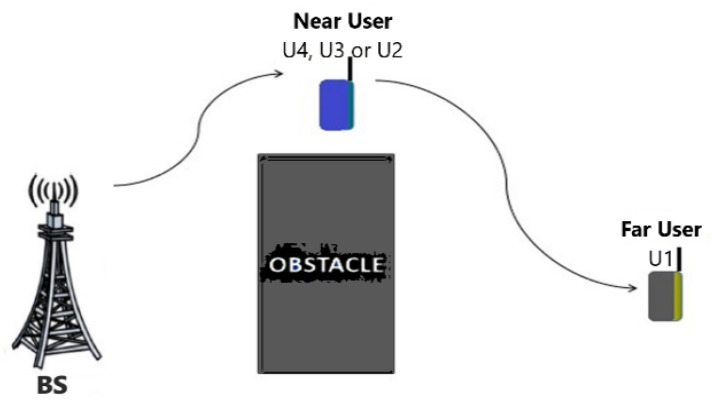
Cooperative SWIPT hybrid THz NOMA model.

**Figure 5 sensors-22-06200-f005:**
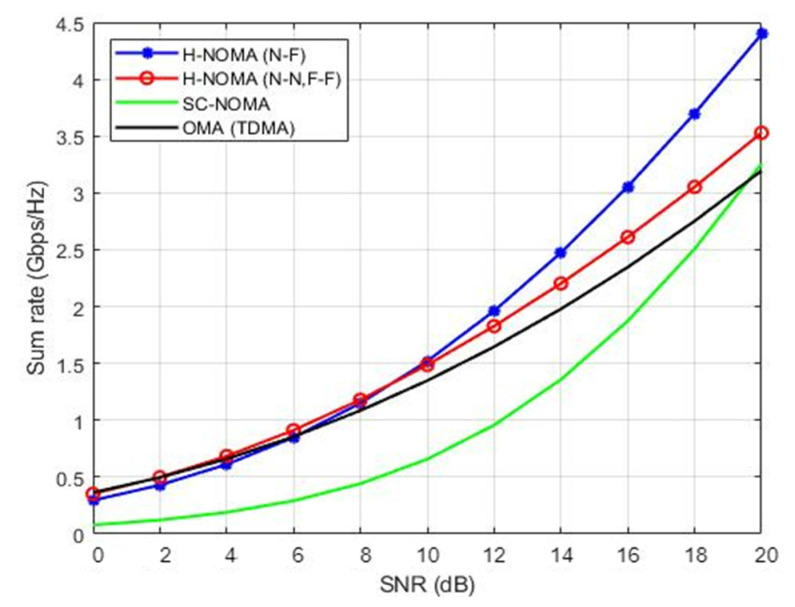
H-NOMA techniques vs. NOMA/OMA.

**Figure 6 sensors-22-06200-f006:**
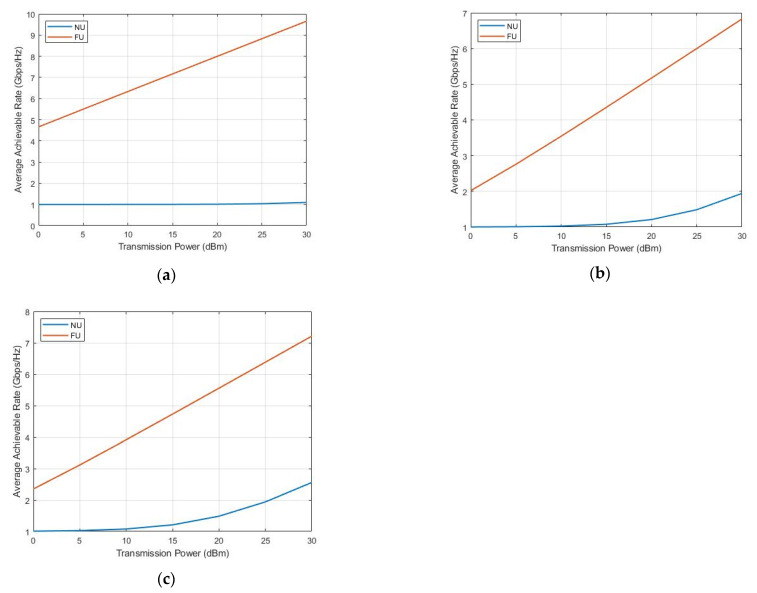
Average rate versus transmission power.

**Figure 7 sensors-22-06200-f007:**
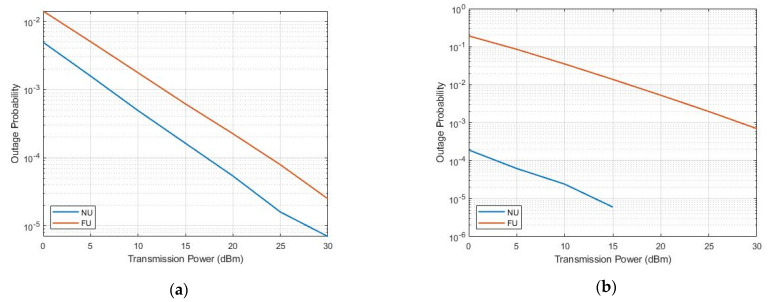
OP versus transmission power.

**Figure 8 sensors-22-06200-f008:**
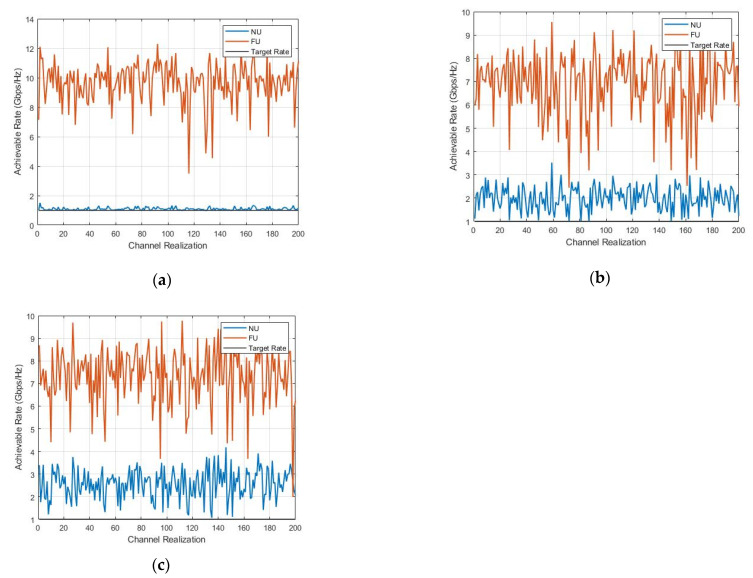
Instantaneous-rate versus channel-realization.

**Figure 9 sensors-22-06200-f009:**
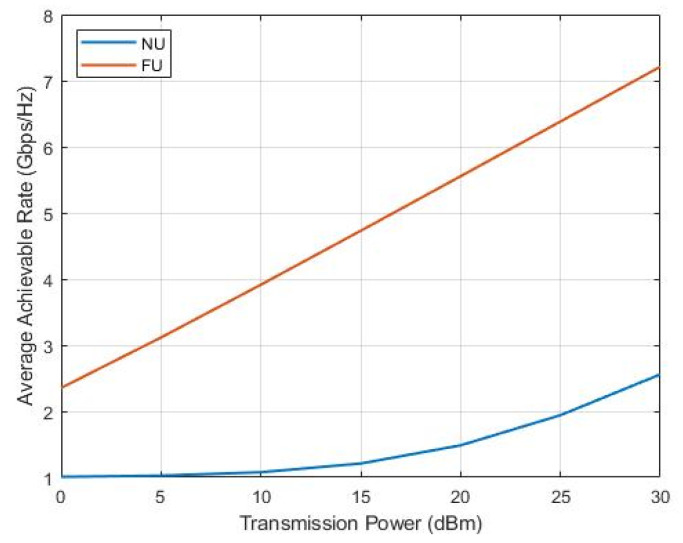
Average rate versus transmission power.

**Figure 10 sensors-22-06200-f010:**
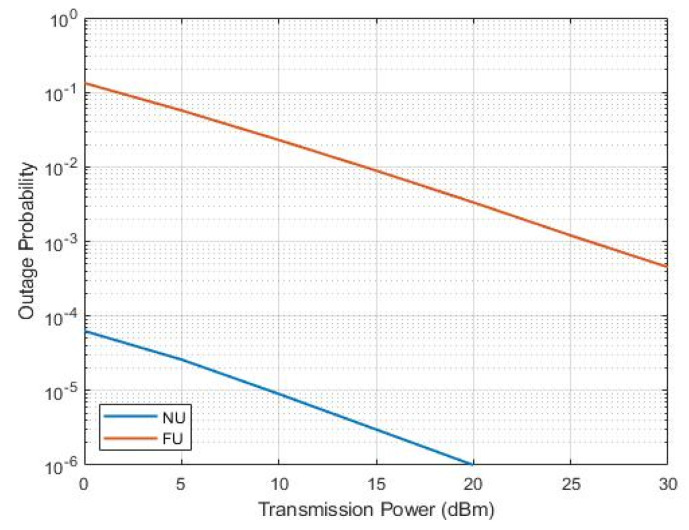
OP versus transmission power.

**Figure 11 sensors-22-06200-f011:**
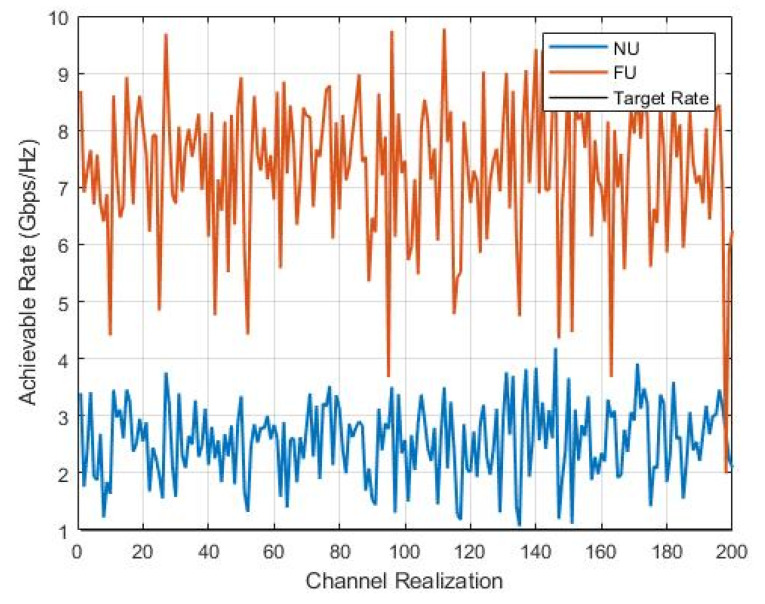
Instantaneous-rate versus channel realization.

**Figure 12 sensors-22-06200-f012:**
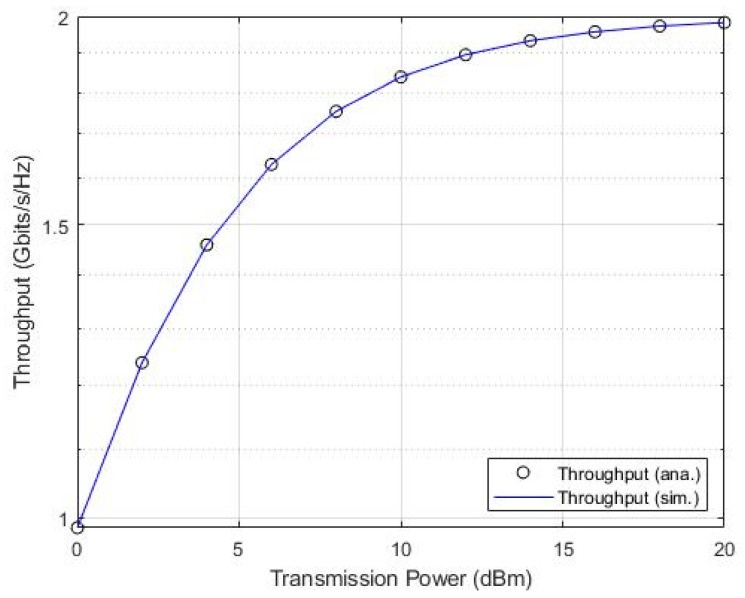
Sum throughput versus transmission power.

**Figure 13 sensors-22-06200-f013:**
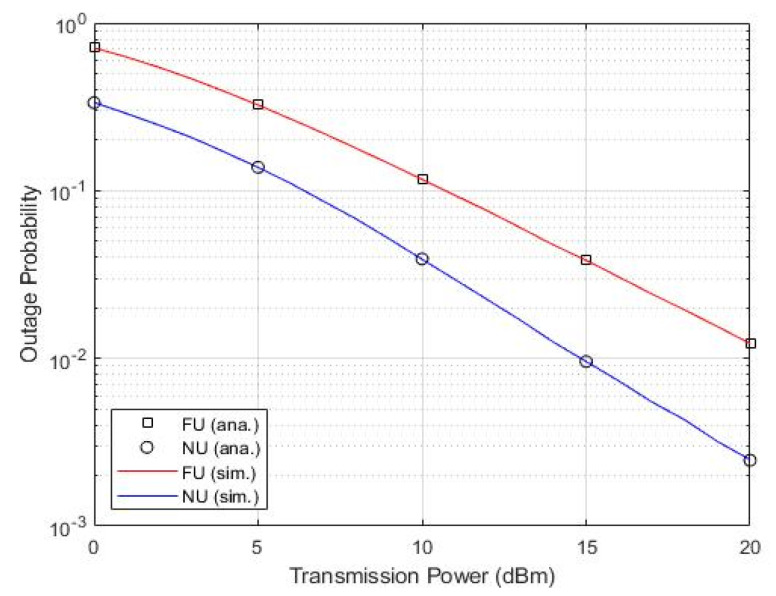
User OP versus transmission power.

**Table 1 sensors-22-06200-t001:** Mathematical equation symbols.

Symbols	Description
AWGN	Additive white Gaussian noise
$\sigma^{2}$	AWGN variance
PDF	Probability density function

P	Transmit power
*αn*	Power coefficient of the near user
*αf*	Power coefficient of the far user
NU	Near user
FU	Far user
xn	Signal of the near user
xf	Signal of the far user
R	Relaying user
H	A vector of Rayleigh channels for k-number of users
K	Constant
N–N	Near–near
F–F	Far–far
*h*2, *h*3, and *h*4	Rayleigh fading channels of user2, user3, and user4
A	Number of antennas
G	Antenna-gain
η	THz source-to-destination losses
*f*	Frequency
*d*	Source-to-destination distance
*a*(*f*)	Absorption coefficient
*c*	Speed of light
*hsn*	BS-to-NU Rayleigh channel (mean = 0, variance = dsn^−η^)
dsn	BS-to-NU transmission distance
*wn*	AWGN
*weh*	Thermal noise (zero mean, variance = $\sigma^{2}$)
ζ	Electronic circuits’ EH efficiency
hnf	NU–FU Rayleigh fading channel
δ	A very small value of $10^{-6}$
ψ	The energy harvesting power fraction

**Table 2 sensors-22-06200-t002:** Simulation parameters.

Symbol	Parameter	V.A	V.B
f	Frequency	311.04 GHz	311.04 GHz
BW	Bandwidth	12.96 GHz	12.96 GHz
P	Transmission power	20–40 dBm	30 dBm
d	Transmission distance	U1, U2, U3, U4 = (10, 9, 4, 3) m	
*αn*	NU power coefficient	0.2 of total power	
*αf*	FU power coefficient	0.8 of total power	
G	Antenna gain	25 dB	
eta	Path loss exponent	4	
	Targeted data rate	1 Gbps	
	EH conversion efficiency	0.7	

**Table 3 sensors-22-06200-t003:** Hybrid-NOMA strategies.

Users/Time	Multiple Access Technique	Strategy				Time Slots
**U1, U2, U3, and U4**	TDMA	U1	U2	U3	U4	4
	NOMA	U1, U2, U3, and U4				1
	NOMA\TDMA	First Pair		Second Pair		2
Time	4 ms					

## Data Availability

Not applicable.

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
