# Peer review of "SWIPT-Pairing Mechanism for Channel-Aware Cooperative H-NOMA in 6G Terahertz Communications"

_sensors, 2022, doi:10.3390/s22166200_

Round 1

Reviewer 1 Report

In this work, the authors propose integrating cooperative simultaneous- 10 ous wireless information and power transfer (SWIPT) and hybrid- non-orthogonal multiple access 11 (H-NOMA) using THz frequency bands. The authors have proposed a novel method that improves energy and spectral efficiencies significantly. However, there are some major concerns that need to be addressed before the paper can be accepted for publication. The concerns are as follows:

1. The quality of figure 2 should be improved.

2. For better readability, a notation table should be included to describe important mathematical symbols.

3. Figure 5 is blurry.  Improve the quality of fig. 5

4. Write more discussion and analysis for the results obtained in fig 8 (a), (b), (c), and fig. 10, fig. 11,  fig 13, and fig 14.

5.  The literature survey should improve by including more recent papers and high cited journals such as the following work:

(2021) "Secrecy Rate Maximization in Virtual-MIMO Enabled SWIPT for 5G Centric IoT Applications," in IEEE Systems Journal, vol. 15, no. 2, pp. 2810-2821

6. Include the future work in the conclusion as well.

Author Response

Thank you for the valuable comments,

All is done, in addition to the manuscript's entire revision.

Reviewer 2 Report

1. Line 315: From (17) not (27)!

2. Figure 6: SNR starts with 20dB value? I think it's not significant!!! Reproduce the figure with 0dB! H-NOMA (N-F), H-NOMA (N-N, F-F) and OMA (TDMA) give the same results between 20dB and 28dB, who do you explain this?

3. Figure 7: This figure is not clear! Reproduce this figure!

4. The same remarque for Figure 8!

5. Line 355: Why you use just BPSK and QPSK modulations? The high throughput requires more order on modulations like 16-QAM, 64-QAM, ...

Author Response

Thank you for the valuable comments,

All was done, in addition, an entire revision was done to the manuscript.

Round 2

Reviewer 1 Report

Authors addressed all the comments. I recommend to accept the paper.